# The Effects of Optimal Dietary Vitamin D_3_ on Growth and Carcass Performance, Tibia Traits, Meat Quality, and Intestinal Morphology of Chinese Yellow-Feathered Broiler Chickens

**DOI:** 10.3390/ani14060920

**Published:** 2024-03-16

**Authors:** Junjie Wei, Ling Li, Yunzhi Peng, Junyi Luo, Ting Chen, Qianyun Xi, Yongliang Zhang, Jiajie Sun

**Affiliations:** 1Guangdong Provincial Key Laboratory of Animal Nutrition Control, College of Animal Science, South China Agricultural University, Guangzhou 510642, China; 2233139108@stu.scau.edu.cn (J.W.); 15014226638@163.com (L.L.); luojunyi@scau.edu.cn (J.L.); allinchen@scau.edu.cn (T.C.); xqy0228@163.com (Q.X.); 2Ministry of Agriculture Key Laboratory of Animal Nutrition and Feed Science, Department of Poultry Nutrition and Feed Science, WENS Research Institute (Technology Center), Guangdong Wens Foodstuff Group Co., Ltd., Yunfu 527400, China; 18385642757@163.com

**Keywords:** vitamin D_3_, yellow-feathered broilers, production performance, intestinal microorganisms, liver transcriptome

## Abstract

**Simple Summary:**

Vitamin D_3_ is an essential trace element in the poultry diet and plays an important role in healthy poultry production. Establishing the amount of VD_3_ required in the yellow feather broiler diet is important for actual production. This experiment aims to determine the appropriate amount of VD_3_ for enhancing growth performance, tibia traits, slaughter performance, and meat quality and to further explore the metabolic pattern of VD_3_ in broilers by 16S rRNA and liver transcriptome sequencing.

**Abstract:**

This study aimed to assess the effects of different dietary vitamin D_3_ (VD_3_) levels on growth and carcass performance, tibia traits, meat quality, and intestinal morphology of yellow-feathered broilers. One-day-old broilers (n = 1440) were assigned into four treatment groups with six replicates per group, and each replicate contained 60 chicks. Dietary VD_3_ significantly improved the growth performance and carcass traits of broilers, and only low-dose VD_3_ supplementation decreased the abdominal fat percentage. High-dose VD_3_ supplementation improved intestinal morphology in the finisher stage, while the b* value of breast muscle meat color decreased markedly under VD_3_ supplementation (*p* < 0.05). Serum Ca and P levels and the tibia composition correlated positively with dietary VD_3_ supplementation at the early growth stage. The weight, length, and ash contents of the tibia increased linearly with increasing dietary VD_3_, with maximum values achieved in the high-dose group at all three stages. Intestinal 16S rRNA sequencing and liver transcriptome analysis showed that dietary VD_3_ might represent an effective treatment in poultry production by regulating lipid and immune-related metabolism in the gut–liver axis, which promotes the metabolism through the absorption of calcium and phosphorus in the intestine and improves their protective humoral immunity and reduces infection mortality. Dietary VD_3_ positively affected the growth—immunity and bone development of broilers during the early stage, suggesting strategies to optimize poultry feeding.

## 1. Introduction

Vitamin D_3_ (VD_3_), the main form of vitamin D, is a critical nutrient for maintaining normal life activities in animals. VD_3_ plays a crucial role in the absorption and metabolism of calcium and phosphorus [1]. In animals, VD_3_ is produced in the skin by exposure to sunlight or obtained through dietary supplements [2]. In commercial poultry farming, the inclusion of VD_3_ in their diets is essential to ensure optimal growth and normal development. This is very important because poultry often have deficient access to sunlight under intensive farming conditions [3]. Poultry is a diurnal animal whose visual system is very sensitive to light. Light is one of the important environmental factors in livestock and poultry production, and it has an important impact on the growth, development, and productivity of poultry.

VD_3_ is transported to the liver, where it is hydroxylated to 25-hydroxycholecalciferol (25-OH-D_3_) by 25-hydroxylase in the hepatocyte mitochondria [4]. Subsequently, in the proximal tubular epithelial cells of the kidney, 25-OH-D_3_ is further converted to the active form, 1,25-dihydroxycholecalciferol (1,25-(OH)_2_-D_3_) by 1α-hydroxylase [4]. VD_3_ and its derivatives are believed to stimulate calcium and phosphate absorption in the gut and directly or indirectly regulate bone formation and reabsorption [5]. Increasing levels of VD_3_ in their diet significantly reduced the incidence of tibial dyschondroplasia and field rickets in broilers [6]. At the same time, incubated chicks fed a higher VD_3_ diet exhibited higher tibial ash compared with those fed a lower VD_3_ diet [7]. Previous studies also demonstrated that VD_3_ plays a vital role in poultry production by improving daily weight gain, feed intake, and the feed conversion ratio [6]. Moreover, VD_3_ metabolites have been demonstrated to reduce the inflammatory response, maintain intestinal health, enhance immune function [8], regulate energy metabolism [9], and improve the meat quality of broilers [6]. The beneficial role of VD_3_ in growth and development might be linked to the gut microbiome [10]. VD_3_ is likely to influence the composition of the gut microbiome by modulating immune and inflammatory responses [11].

In commercial broiler farming models, the VD_3_ content in the diet is commonly set at 10 to 20 times higher than the recommendation provided by the National Research Council (NRC). This much greater level of provision reduces the incidence of field rickets and tibial dyschondroplasia [6]. In contrast, yellow-feathered broilers require a lower level of VD_3_ during their growth and fattening periods compared with that required at the chick stage [12]. Therefore, this study aimed to investigate the suitable supplemental level of VD_3_ for yellow-feathered broilers at different growth stages by examining their growth performance, tibia traits, slaughter performance, and meat quality. To gain further insights into the potential mechanisms of VD_3_ in yellow-feathered broilers, we adopted a combined approach involving cecal microbiota 16S sequencing and liver transcriptome analysis. The hypothesis of the study was that VD_3_ supplementation would improve the growth performance, tibia traits, slaughter performance, and meat quality of yellow-feathered broilers.

## 2. Materials and Methods

### 2.1. Animals, Experimental Design, and Treatments

The study was conducted at a large-scale chicken farm in Yunfu, Guangdong, China. A total of 1440 yellow-feathered broilers (one day old and approximately 32 g per chick) were randomly assigned to four groups with six replicates per group, and each replicate contained 60 chicks. At the starter stage (days 1 to 21), the diets of the control (CN) group, low-concentration (LC) group, medium-concentration (MC) group, and high-concentration (HC) group were supplemented with 0, 1200, 2400, and 3600 IU/kg of VD_3_, respectively. At the grower stage (days 22 to 42) and finisher stage (days 43 to 63), the diets of the CN group, LC group, MC group, and HC group were supplemented with 0, 1000, 2000, and 3000 IU/kg of VD_3_, respectively. The basic composition and nutritional level of the broiler feed is shown in Appendix A. The birds were housed in rectangular pens with a stocking density of 0.09 m^2^ per bird. Each pen was equipped with an automatic feeding bucket and automatic drinking system, and rice husks were provided as bedding to a depth of 5 cm. The temperature and lighting conditions in the pens were monitored daily and adjusted as required, following the rules of the chicken farm. The experiment lasted for 63 d, and feed and water were offered ad libitum to the birds. All birds were vaccinated against viral diseases, including Marek’s disease, Newcastle disease, infectious bursal disease, and fowl pox, according to the recommended schedule. The Animal Care Committee at South China Agricultural University approved all the animal procedures used in this study.

### 2.2. Measurements and Sampling

Mortality was monitored daily, and any deceased birds were weighed and used to adjust for feed intake. Feed consumption was recorded weekly, and the total weight per pen was measured at 1, 21, 42, and 63 d to calculate the average daily feed intake (ADFI), average daily gain (ADG), and feed conversion ratio (FCR). On the morning of days 21, 42, and 63, after a 12-h feed withdrawal period, two birds in each pen (i.e., 12 birds per group) were selected for sampling. Blood samples of approximately 3 mL were collected from the wing vein, immediately placed on ice for more than 1 h, and then centrifuged at 3000× *g* at 4 °C for 10 min. The serum was then stored at −20 °C for further analysis. After blood sampling, the birds were euthanized using CO_2_ asphyxiation. Samples from the middle fragments (1 cm) of the duodenum were taken and preserved in a 4% paraformaldehyde solution for subsequent hematoxylin and eosin (H&E) staining and microscopic measurement. The tibias were removed and frozen at −20 °C for analyses of bone characteristics and composition.

On the last day of the experiment (day 63), various measurements were taken after each bird was slaughtered. The following parameters were recorded: dressed weight, eviscerated yield, dressing percentage, eviscerated yield percentage, breast muscle yield percentage, thigh muscle yield percentage, and abdominal fat percentage. All yield calculations were conducted according to the “Terminology of Poultry Production Performance and Statistical Method of Measurement (NY/T823-2020, China)”. Within 10 min postmortem, all breast and thigh muscles were harvested and stored at 4 °C for subsequent measurement of meat quality. Additionally, the liver and cecal contents were collected immediately, snap-frozen in liquid nitrogen, and then stored at −80 °C until further processing.

### 2.3. Serum Biochemical Determinations

Total protein, albumin, globulin, glucose, aspartate aminotransferase (AST), creatinine, urea, uric acid, total cholesterol, high-density lipoprotein cholesterol (HDL-C), low-density lipoprotein cholesterol (LDL-C), and triglyceride were detected using an automatic biochemical analyzer (Hitachi 902 Automatic Analyzer, Tokyo, Japan). Total calcium and dialysable phosphorus in the serum were determined spectrophotometrically employing assay kits (Jiancheng Bioengineering Institute, Nanjing, China).

### 2.4. Intestinal Histomorphological Measurements

The intestinal tissue was processed following the procedure described in Sakkas et al. [13]. After fixation in 4% paraformaldehyde for 48 h, the intestinal tissues were washed using 0.01 M phosphate-buffered saline (PBS), dehydrated using increasing concentrations of ethanol and cleaned with xylene. Subsequently, each tissue was embedded in paraffin wax, cut into 5 μm thick slices, fixed on slides, and gradually stained with H&E. The slides were washed with distilled water for 5 min, stained with hematoxylin for 6 min, washed with tap water, treated with a differentiated solution for 20 s, soaked in water for 20 min to blue, stained with 1% eosin for 1 min, rinsed in tap water, dehydrated, made transparent, and sealed. The stained slides were then examined under an optical microscope (OLYMPUS IX71, Olympus, Tokyo, Japan), and the villus length and crypt depths of the stained sections were measured using custom-written scripts in Image J software version 1.8.0 (National Institutes of Health, Bethesda, MD, USA).

### 2.5. Bone Characteristics

The bone characteristics were analyzed using the method described by Cromwell et al. [14]. After thawing the tibias overnight at 4 °C, any adhering soft tissues were removed. The surface of the bone was then dried using blotting paper. The weight of each bone was measured using an analytical balance, and the length was measured using Vernier calipers. Subsequently, these bone samples were soaked in anhydrous ethanol for 24 h and anhydrous ethyl ether for 48 h. They were then oven-dried at 105 °C for 24 h to determine the dry defatted bone weight. Thereafter, the samples were crushed and incubated at 550 °C for 24 h inside a muffle furnace to calculate the ash content based on the percentage of dry-defatted weight. The ash was dissolved by heating with nitric and hydrochloric acid, and then the calcium content was determined using ethylenediaminetetraacetic acid (EDTA) titration. Additionally, the phosphorus content was determined using multifunctional enzyme markers (Bio Tek, Winooski, VT, USA) based on molybdenum yellow colorimetry.

### 2.6. Meat Quality

At 45 min (pH45min) and 24 h (pH24h) after slaughter, the pH values of the breast and thigh samples were measured using a portable pH meter (Testo 206, Testo, Lenzkirch, Germany). Before taking the measurements, the pH meter was calibrated using buffers with pH values of 4.01, 7.00, and 10.01 to ensure accuracy. At 45 min after slaughter, the samples, including lightness (L*), redness (a*), and yellowness (b*), were assessed using a Minolta Chroma Meter (CR-300, Konica Minolta, Dietikon, Switzerland). These measurements represent the color attributes of the meat, among which the most commonly used is the L* value; the smaller the value, the more reddish the meat color. Additionally, the drip loss of the samples was calculated as the percentage of weight loss after being suspended at 4 °C for 24 h.

### 2.7. 16S rRNA Sequencing and Annotation Analysis of Cecal Microorganisms

Based on the findings mentioned above, we found that the LC treatment did not have any significant impact on the growth performance of yellow-feathered broilers. The results indicated that the LC treatment might provide more economic benefits than the other VD_3_ supplementation levels. As a result, the LC and the CN groups were selected for 16S rRNA sequencing and RNA sequencing (RNA-Seq) experiments.

Total genomic DNA was extracted from the cecal digesta, and the integrity and concentration of the genomic DNA were assessed using 1% agarose gel electrophoresis and a Qubit Fluorometer (Invitrogen, Carlsbad, CA, USA), respectively. The DNA was then diluted to 1 ng/μL using sterile water. Next, the distinct V3–V4 regions of the 16S rRNA genes were amplified using polymerase chain reaction (PCR) with specific primers that included barcodes (forward: CCTAYGGGRBGCASCAG; reverse: GGACTACNNGGGTATCTAAT). To prepare the sequencing library, the Illumina TruSeq^®^ DNA PCR-Free Sample Preparation Kit was used (Illumina Inc., San Diego, CA, USA). The prepared library was assessed for quality and quantity before being sequenced on the NovaSeq6000 platform (Illumina Inc.). The process of obtaining reads involved two steps: strict quality filtering and chimeric sequence removal [15]. After these steps, the effective tags that showed ≥97% similarity were grouped into one operational taxonomic unit (OTU) [16]. The Mothur algorithm was used to annotate the feature sequences with taxonomic information, utilizing the Silva 132 database [17]. All the samples were randomly subsampled for normalization, and subsequent analysis of alpha diversity and beta diversity was conducted based on the normalized data.

### 2.8. RNA-Seq and Functional Annotation

Total RNA was extracted from the liver employing the Trizol reagent (Invitrogen) according to the manufacturer’s recommendations. RNA quality was assessed using an Agilent 2100 Bioanalyzer (Agilent Technologies, Santa Clara, CA, USA) and checked using RNase-free agarose gel electrophoresis. The RNA-seq library was constructed using an NEB Next Ultra RNA Library Prep Kit for Illumina (NEB#7530, New England Biolabs, Ipswich, MA, USA). Then, the library was sequenced on an Illumina Novaseq6000 sequencer.

To obtain high-quality clean reads, the low-quality reads were filtered out using fastp v0.18.0 [18], and the rRNA was further removed using the short-reads alignment tool Bowtie2 v2.2.8 [19]. The paired-end clean reads were mapped to the reference genome using HISAT2 v2.4.0 [20]. Thereafter, the mapped reads of each sample were assembled using StringTie v1.3.1 [21] in a reference-based approach. For each transcribed region, an FPKM (fragment per kilobase of transcript per million mapped reads) value was calculated to quantify expression abundance [22]. The differentially expressed genes (DEGs) between the two groups were identified using DESeq2 v.1.1.25 [23]. The genes with a *p* value below 0.05 and an absolute fold change ≥2 were considered as DEGs. The DEGs were annotated to pathways in the Kyoto Encyclopedia of Genes and Genomes (KEGG) employing the Database for Annotation, Visualization, and Integrated Discovery (DAVID) v6.8 [24]. Moreover, we performed Gene Set-Enrichment Analysis (GSEA) using GSEA v4.0.2 [25] to identify whether a set of genes in specific KEGG pathways showed significant differences between the two groups.

### 2.9. Statistical Analysis

One-way analysis of variance (ANOVA) followed by a least significant difference (LSD) multiple comparison test was used to evaluate the differences among the treatment groups employing SPSS software 17.0 (IBM Corp, Armonk, NY, USA). The results were expressed as the mean ± standard error (SE), and a value of *p* < 0.05 was considered statistically significant.

## 3. Results

### 3.1. Growth Performance and Biochemical Analyses

The results of growth performance are presented in Table 1. Compared with the CN group, the body weight (BW) of the broilers at 63 days in the three experimental groups was significantly increased (*p* < 0.05), and no significant differences were found among the different treatment groups (*p* > 0.05). From day 1 to 63, the ADFI and ADG significantly increased in the LC, MC, and HC groups compared to the CN group (*p* < 0.05), while the FCR significantly decreased (*p* < 0.05). From day 22 to 63, the ADFI, ADG, and FCR did not exhibit significant differences among the three experimental groups (*p* > 0.05), and only the FCR in the LC group was significantly lower than that in the HC group (*p* < 0.05). In addition, supplementation with VD_3_ had no substantial impact on the measured serum biochemical indices (*p* > 0.05) (Appendix A).

### 3.2. Intestinal Morphology

Dietary supplementation with VD_3_ had a significant impact on villus length (VL) and crypt depth (CD) (Table 2). In detail, compared with those in the CN group, the VL and V/C ratio in the MC and HC groups had increased significantly by day 63 (*p* < 0.05). In contrast, there were no significant differences between the CN and LC groups (*p* > 0.05). Specifically, on day 63, the VL and V/C ratios were significantly higher in the MC and HC groups compared to the CN group (*p* < 0.05). Conversely, no significant differences were observed between the CN and LC groups (*p* > 0.05).

### 3.3. Serum Ca, P, and Tibial Characteristics

Serum calcium concentrations in broilers at day 21 were significantly elevated in the MC and HC groups compared with those in the CN and LC groups (*p* < 0.05). The serum dialyzable P was highest in the HC group at day 42 compared with that in the other three groups (*p* < 0.05) (Appendix A). The results showed that the serum Ca and P levels correlated positively with dietary VD_3_ supplementation. As depicted in Table 3, the weight, length, and ash composition of the tibia increased in response to increasing dietary VD_3_. At days 42 and 63, tibia weight, length, and ash contents had increased significantly in the LC, MC, and HC groups compared with those in the CN group (*p* < 0.05). In contrast, no significant differences were observed among the three VD_3_ treatment groups (*p* > 0.05). At day 21, the HC group had the highest weight, length, and ash contents of the tibia, and these measurements were significantly higher than those in the CN group (*p* < 0.05). The contents of Ca and P in the tibia of the MC group were maximized and significantly higher than those in the CN group at days 21 and 42 (*p* < 0.05). However, no significant differences in Ca and P levels in the tibia were observed among the groups at day 63 (*p* > 0.05).

### 3.4. Carcass Characteristics and Meat Quality

The relevant indices of carcass characteristics are presented in Appendix A. VD_3_ supplementation in the LC, MC, and HC groups resulted in an increased live weight at slaughter, dressed weight, and eviscerated yield of the broilers (*p* < 0.05), and there were no significant differences among the three VD_3_ supplementation groups (*p* > 0.05). Dietary supplementation of VD_3_ significantly increased the breast muscle yield percentage in comparison to that in the CN group (*p* < 0.05); however, no significant differences were observed in the thigh muscle yield percentage among the different groups (*p* > 0.05). Compared with the CN group, the abdominal fat percentage in the LC group was significantly decreased (*p* < 0.05); however, no significant differences were found in the MC and HC groups in comparison with the CN group (*p* >0.05). As indicated in Appendix A, dietary VD_3_ had no effect on meat color (L*, a*), initial pH (pH_45min_), pH at 24 h (pH_24h_), and the drip loss in breast and thigh muscles (*p* > 0.05). The meat color b* value of the breast muscle decreased significantly in three VD_3_ supplementation groups compared with that in the CN group (*p* < 0.05). Regarding the drip and cooking loss, no significant effects of dietary VD_3_ were observed between the VD_3_ supplementation and CN groups (*p* > 0.05).

### 3.5. 16S rRNA Gene Sequencing and Annotation Analysis

An average of 80,524 ± 947 raw reads in each sample were generated through high-throughput sequencing analysis. Following quality control, an average of 83.49 ± 1.55% of the valid reads from each sample were retained for further analysis (Appendix A). We identified 7595 OTUs in the cecal contents of yellow-feathered broilers in the CN and LC groups. Among them, 2053 OTUs were present in both groups, and these OTUs were therefore defined as the core OTUs (Figure 1A). The top 10 phyla in the core OTUs were Firmicutes, Bacteroidota, Proteobacteria, Cyanobacteria, Actinobacteriota, Campilobacterota, Acidobacteriota, Verrucomicrobiota, Desulfobacterota, and Nitrospirota (Appendix A; Figure 1B). At the genus level, the dominant microbiota were *Clostridia_UCG-014*, *Alistipes*, *Bacteroides*, *Faecalibacterium*, *Ruminococcus_torques_group*, *Clostridia_vadinBB60_group*, *Lactobacillus, Blautia*, *Parabacteroides* and *Escherichia-Shigella* (Figure 1C). The Chao1, Goods_coverage, Observed species, Shannon, and Simpson indices between the two groups were calculated, and no significant effects on the alpha diversity of microorganisms in the cecum of broilers were found (*p* > 0.5) (Appendix A). For the beta-diversity, the unweighted (Figure 2A) and weighted (Figure 2B) principal component analysis (PCA) showed that the CN and LC samples could not be separated absolutely, strongly suggesting that no significant differences were observed in response to VD_3_ supplementation at a low concentration. Through statistical analysis of metagenomic profiles (STAMP) analysis, a few genera with significant differences were identified between the two groups (Figure 2C). In detail, the relative abundances of *Megasphaera* and *DNF00809* in the LC group were significantly increased compared with those in the CN group (*p* < 0.05). In contrast, the relative abundances of *Candidatus_Soleaferrea* and *Ruminococcus_gnavus_group* in the LC group were significantly decreased (*p* < 0.05) (Appendix A). The linear discriminant analysis effect size (LefSe) algorithm was employed to identify statistically significant biomarkers among the two groups, with a Linear Discriminant Analysis (LDA) score higher than two. A total of seven genera could be regarded as biomarkers for CN samples, including *Anaerotruncus*, *Rikenellaceae_RC9_gut_group*, *Marvinbryantia*, *Treponema*, *Candidatus_Soleaferrea*, *Bacteroidales_RF16_group*, and *Ruminococcus_gnavus_group*. In comparison, seven genera could be regarded as biomarkers for the LC samples, including *Akkermansia*, *Megasphaera*, *Tuzzerella*, *Ezakiella*, *Coriobacteriaceae_UCG_002*, *Arcanobacterium*, and *DNF00809* (Appendix A).

To determine the relative contribution of the identified genera to the compositional dissimilarity between different groups, a similarity percentage (Simper) analysis was performed [26]. The results determined that *Clostridia_UCG-014* (10.16% of community dissimilarity), *Alistipes* (7.29%), *Bacteroides* (6.75%), *Faecalibacterium* (6.61%), and *Lactobacillus* (2.98%) contributed the most to the dissimilarity in the active community in response to VD_3_ supplementation (Figure 2D). Pearson correlation analysis was conducted to investigate the relationship between the top 10 contributing genera and the phenotypic characteristics of broilers at day 63 (Appendix A). The relative abundance of *Clostridia_UCG-014* was found to correlate negatively with the content of tibial Ca (*p* < 0.05), while *Escherichia-Shigella* correlated positively with the content of tibial Ca (*p* < 0.05). In addition, *Bacteroides* correlated positively with the drip loss of breast muscle (*p* < 0.05). The relative abundance of *Faecalibacterium* correlated strongly and positively with tibia weight (*p* < 0.01) and correlated negatively with the pH_45min_ of breast muscle (*p* < 0.01). The relative abundance of *Lactobacillus* correlated negatively with tibial P and flesh color a* (*p* < 0.05). Furthermore, the genus *Ruminococcus_torques_group* correlated positively with eviscerated yield (*p* < 0.05), while *Blautia* correlated negatively with the pH_45min_ of breast muscle (*p* < 0.01).

### 3.6. Liver Transcriptome Analysis

Three cDNA libraries were constructed from each group, and the average number of clean reads per library was 48,740,205 ± 1,162,941, with an average retention rate of 99.41 ± 0.34% after removing low-quality reads (Appendix A). The average unique mapping rate and overall mapping rate were 93.0 ± 0.16% and 95.69 ± 0.18%, respectively (Appendix A). Moreover, approximately 92.02 ± 0.16% of the reads were distributed in the exon region (Appendix A). These findings validated the successful construction of the libraries and their suitability for subsequent analyses. In total, 17,827 known transcripts were identified across all six libraries, and each transcript abundance was quantified using FPKM values (Appendix A). We also performed PCA using the FPKM values of the identified candidates. The results showed that CN and LC individuals could be separated absolutely into two groups, especially in LC groups, which reflected the differences in gene expression in response to dietary VD_3_ supplementation (Figure 3A). In comparison with the CN libraries, we identified a total of 412 significantly differentially expressed genes (DEGs) (Appendix A), including 138 up- and 274 downregulated genes in the LC libraries (Figure 3B). The DEGs were found to participate in 181 KEGG signaling pathways (Appendix A), mainly including a number of lipid and immune-related metabolic terms, such as the peroxisome proliferator-activated receptor (PPAR) signaling pathway, immune network for IgA production, and steroid hormone biosynthesis and fatty acid metabolism (Figure 3C). Then, KEGG pathway genes were subjected to gene set enrichment analysis (GSEA), and several genes in 21 signaling pathways showed significant enrichment (Appendix A), such as Th17 cell differentiation and the PPAR signaling pathway (Figure 3D).

## 4. Discussion

VD_3_ is an essential trace element in poultry diets and plays a vital role in green, healthy poultry production. The minimum recommended dose of VD_3_ in broiler diets, as prescribed by the NRC, is 200 IU/kg [27]. In commercial broiler diets, VD_3_ is typically provided at higher levels than the NRC recommendation; however, no accurate dose for yellow-feathered broiler chickens has been proposed [28]. In the present study, the growth performance of the birds revealed a significant treatment effect by dietary VD_3_ supplementation during the whole experiment. Similar to a previous study [6], dietary VD_3_ significantly increased the ADFI and ADG from 1 to 42 days and improved the FCR of the tested broilers from 43 to 63 days. The improved growth performance induced by VD_3_ supplementation might act by promoting intestinal development and health through improving the status of the development of intestinal villi, promoting the activity of related digestive enzymes, and the intestinal antioxidant capacity, and maintaining the balance of intestinal flora [29]. In general, intestinal morphology is a key factor affecting the digestion and absorption of nutrients in poultry [30], and villus length, crypt depth, and the V/C ratio are significant indicators used to assess the maturity and functional capacity of intestinal tissues [31]. A higher V/C ratio indicates a stronger capacity for nutrient absorption [32]. Numerous previous studies have reported the beneficial effects of VD_3_ metabolites on enteric development. For example, VD_3_ metabolites have been found to promote the formation of intestinal villi [33], upregulate the expression of tight junction proteins, such as Occludin, ZO-1, Claudin-2, and E-cadherin [34], and reduce the intestinal cell apoptosis caused by inflammatory processes [35]. Nutrient absorption in poultry is closely related to intestinal villi morphology. The higher the height of the intestinal villi, the smaller the crypt depth and the greater the V/C value, indicating the stronger the ability to absorb nutrients. Similarly, our findings showed that VD_3_ supplementation in the MC and HC groups increased the duodenal villus length and V/C value and decreased the crypt depth of broilers at day 63. At the same time, better results were obtained with increasing dietary VD_3_ levels. Our observation suggested that a higher level of VD_3_ supplementation is conducive to improving the intestinal development and function of broilers.

In general, VD_3_ has a crucial role in calcium and phosphorus homeostasis and has a vital role in the subsequent regulation of bone characteristics [36]. In its initial form, VD_3_ is not biologically active and must be converted to 25-hydroxyvitamin D_3_ in the liver and subsequently to 1,25-dihydroxyvitamin D_3_ in the kidney. 1,25-dihydroxyvitamin D_3_ mainly acts through a nuclear receptor for calcium and phosphorus absorption in the small intestine [37]. In this study, the serum calcium and phosphorus contents of the birds increased modestly with increasing dietary VD_3_ supplementation, especially in the starter phase (1–21 days) and grower phase (22–42 days), similar to the values obtained in the tibia of broilers. Tibial weight, length, and ash, together with the content of calcium and phosphorus, are commonly used as indicators to evaluate the bone development of poultry [38]. Supplementation with VD_3_ at approximately 3000 IU/kg was optimal for the physical properties and mineral deposition of the tibia in broilers [39]. In line with these results, in the present study, the weight, length, and ash contents of the tibia increased linearly in response to increasing dietary VD_3_, and the maximum values were obtained in the HC group at all three developmental stages. The results implied that feeding with VD_3_ has a favorable effect on calcium and phosphorus absorption and utilization in the early stage, and a high level of dietary VD_3_ is essential and important for the growth and development of broiler tibias at different ages [1].

Previous studies reported that dietary VD_3_ addition had positive effects on animal carcass traits in broiler chickens [40] and pigs [29]. In Wenchang broilers, VD_3_ supplementation reduced weight loss and improved the dressing percentage [40], while dietary VD_3_ tended to decrease the body weight loss of sows during lactation [29]. In agreement, our results showed that VD_3_ supplementation improved carcass traits in the yellow-feathered broilers; however, no significant effect of adding VD_3_ was observed between the three experimental groups, generally indicating that a lower level of dietary VD_3_ was sufficient to improve the carcass performances of broilers. Meat quality is the key criterion of food product evaluation, and a very significant parameter of meat evaluation by consumers is its color, which is perceived as an indicator of meat freshness and quality [41]. In broilers, pigs, and cattle, VD_3_ supplementation was shown to improve meat color [42,43,44]. In the present study, we observed that the VD_3_ supplementation in the LC, MC, and HC groups resulted in a decrease in the b* value in breast muscle. The mechanism by which VD_3_ improves chicken meat color might be its regulation of calmodulin-mediated phosphatase, and the appropriate VD_3_ can reduce the content of malon dialdehyde in the muscle, improve the activity of antioxidant enzymes, improve the antioxidant capacity of the body, and thus improve the meat color [45]. The data presented here indicates that a modest improvement in meat color was obtained with a relatively low level of dietary VD_3_ during the finishing phase of birds.

The composition of the intestinal microbiota might be influenced by VD_3_ through the regulation of immune and inflammatory responses [10,11]. Consistent with previous studies [46,47], Firmicutes and Bacteroidota were the most dominant phyla in the cecum of broilers, and *Faecalibacterium*, *Ruminococcus*, *Lactobacillus*, and *Clostridia* were the main genera of Firmicutes [48,49]. In addition, the alpha and beta diversity of the cecal microbial communities were not significantly affected by dietary VD_3_. This finding was inconsistent with a study conducted in weaned piglets, which suggested that dietary VD_3_ significantly affected the colon microbial beta diversity [50]. This might have been caused by species differences (broilers vs. piglets), tissue differences (cecum vs. colon), or the supplementation level of VD_3_ in the LC group might have been insufficient to cause significant changes in the cecal microbial composition and structure of the tested broilers. At the genus level, *Megasphaera* and *DNF00809* were upregulated, while *Candidatus_Soleaferrea* and *Ruminococcus_gnavus_group* were downregulated in the LC group. A number of studies have indicated that *Megasphaera* contributes to the synthesis of short-chain fatty acids [51], the regulation of intestinal homeostasis [52], and the enhancement of the host immune response [53]. In addition, *DNF00809* belongs to the Eggerthellaceae family, which is responsible for bone formation and differentiation through the microbiota-gut-metabolite-bone axis [54]; it can act as a barrier to prevent invasion by disease-causing microorganisms and influence endocrine organs. A decrease in the abundance of *Candidatus_Soleaferrea* in the LC group was proven to correlate with steatolysis processes [55]. In line with these findings, the abdominal fat content showed a decline with the decreasing abundance of *Candidatus_Soleaferrea* in the cecum. Meanwhile, *Ruminococcus_gnavus_group* is an inflammatory bacterium that promotes the breakdown of intestinal barrier function [56] and induces an inflammatory response [57], which generally agrees with the longer villus length and shallower crypt depth identified in our study. Notably, the relative abundances of *Clostridia_UCG-014*, *Escherichia-Shigella*, *Faecalibacterium*, and *Lactobacillus* correlated significantly with bone characteristics, while *Bacteroides* and *Blautia* correlated strongly with meat quality, and *Ruminococcus_torques_group* was associated with carcass traits. Our findings suggested that the dietary VD_3_ improved the production performance of broilers by changing the abundance of these bacteria and enhancing their intestinal health, subsequent digestion, and absorption of nutrients.

In general, the liver is the main target organ for VD_3_ metabolism [58]. Therefore, the effects of dietary VD_3_ on the liver transcriptome were determined using RNA-seq in our study. KEGG annotation and GSEA analysis of the collected data confirmed that the DEGs were mainly enriched in lipid and immune-related metabolism. In pregnant rats, VD_3_ negatively regulated the expression of various lipogenic genes and enzymes in both adipose and liver tissues, thereby reducing the fat content in pregnant animals [59]. In monocyte-derived macrophages, VD_3_ reduced fatty acid accumulation by regulating proteins involved in lipid transport and clearance [60]. In line with these results, the abdominal fat percentage in the LC group was significantly decreased, and the expression levels of lipid metabolism-associated genes were significantly enriched but downregulated in the PPAR signaling pathway based on the GSEA analysis. Previously, VD_3_ has been linked to immunological processes, and its supplementation might have a role in treating or preventing diseases with underlying autoimmune or pro-inflammatory states [61]. Previously published data revealed that VD_3_ prevents acute liver injury and reduces mortality through its anti-inflammatory and antioxidative activities [62]. In general, VD_3_ acts upon a broad range of immune cells involved in the pathogenesis of diseases, including T cells, dendritic cells, macrophages, and B cells [63]. In agreement with these observations, the immune-related DEGs in our study were significantly associated with Th17 cell differentiation, Th1 and Th2 cell differentiation, the B cell receptor signaling pathway, and the T cell receptor signaling pathway. Our observations suggested that dietary VD_3_ improved the production performance of broilers, probably because of improved lipid and immune-related metabolism.

## 5. Conclusions

In conclusion, supplementation with dietary VD_3_ at different levels had a variety of effects on the growth performances, carcass traits, bone characteristics, meat quality, and intestinal morphology of broiler chickens. Based on our findings, the LC group was optimal, with the optimal dose of 3600/3000 IU/kg. Our findings suggested that VD_3_ might prove to be an effective treatment in poultry production by regulating the lipid and immune-related metabolism in the gut-liver axis. The results highlight the importance of carefully selecting an appropriate VD_3_ supplementation level to achieve the specific desired outcomes in broiler production. Therefore, this study provides valuable insights into optimizing poultry feeding strategies to improve performance and lower production costs.

## Figures and Tables

**Figure 1 animals-14-00920-f001:**
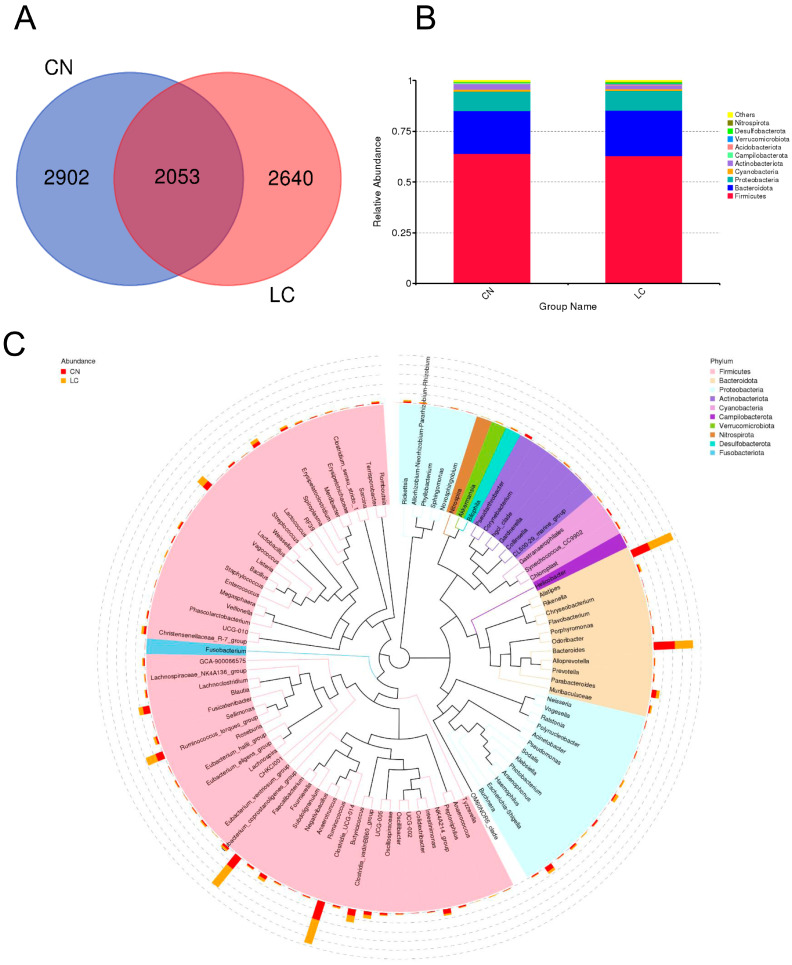
16S ribosomal RNA (rRNA) gene sequencing and annotation analysis. (**A**) A Venn diagram representing the shared and exclusive operational taxonomic units (OTUs) at the 97% similarity level between cecal microbiota in the two groups. (**B**) The bar plot shows the relative abundance of cecal microbiota at the phylum level in each group. (**C**) A phylogenetic tree. Different colors represent different phyla, and different branches represent different genus levels. The closer the distance between the two species, the closer the evolutionary relationship between them. CN, control group; LC, low-concentration group.

**Figure 2 animals-14-00920-f002:**
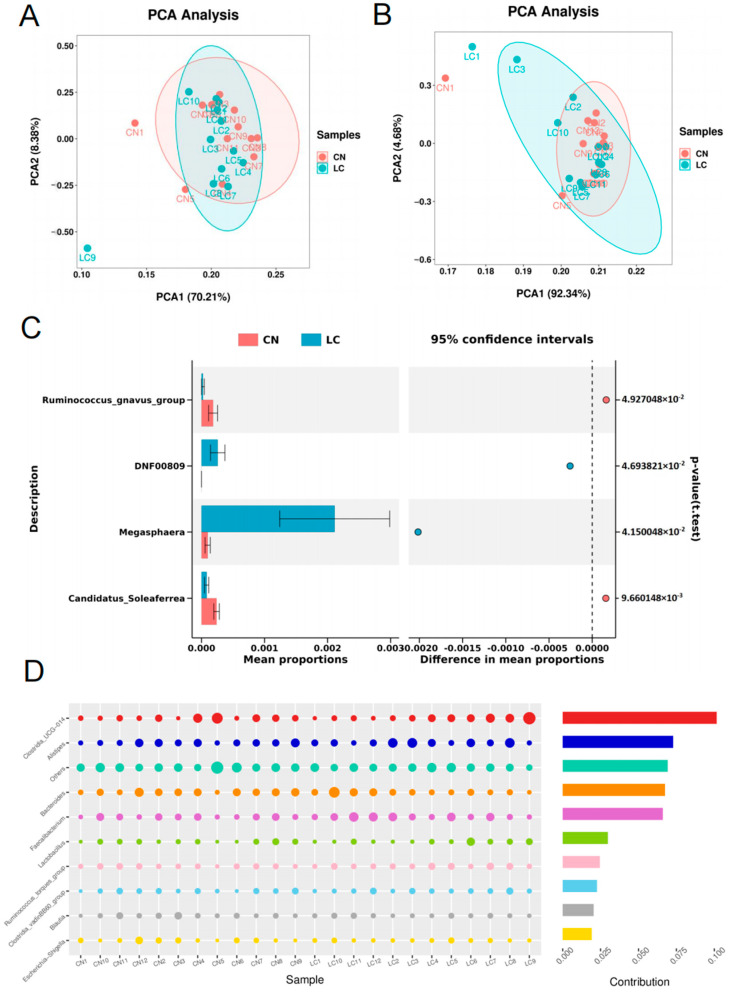
Microbial diversity in the cecum between two groups. (**A**,**B**) Principal component analysis (PCA) at the phylum level with unweighted (**A**) and weighted (**B**) algorithms. (**C**) Statistical analysis of metagenomic profiles (STAMP) shows microorganisms with significant differences at the genus level. (**D**) Similarity percentage (Simper) analysis showing microorganisms that contributed the greatest to dissimilarity in the active community in response to vitamin D_3_ (VD_3_) supplementation. CN, control group; LC, low-concentration group.

**Figure 3 animals-14-00920-f003:**
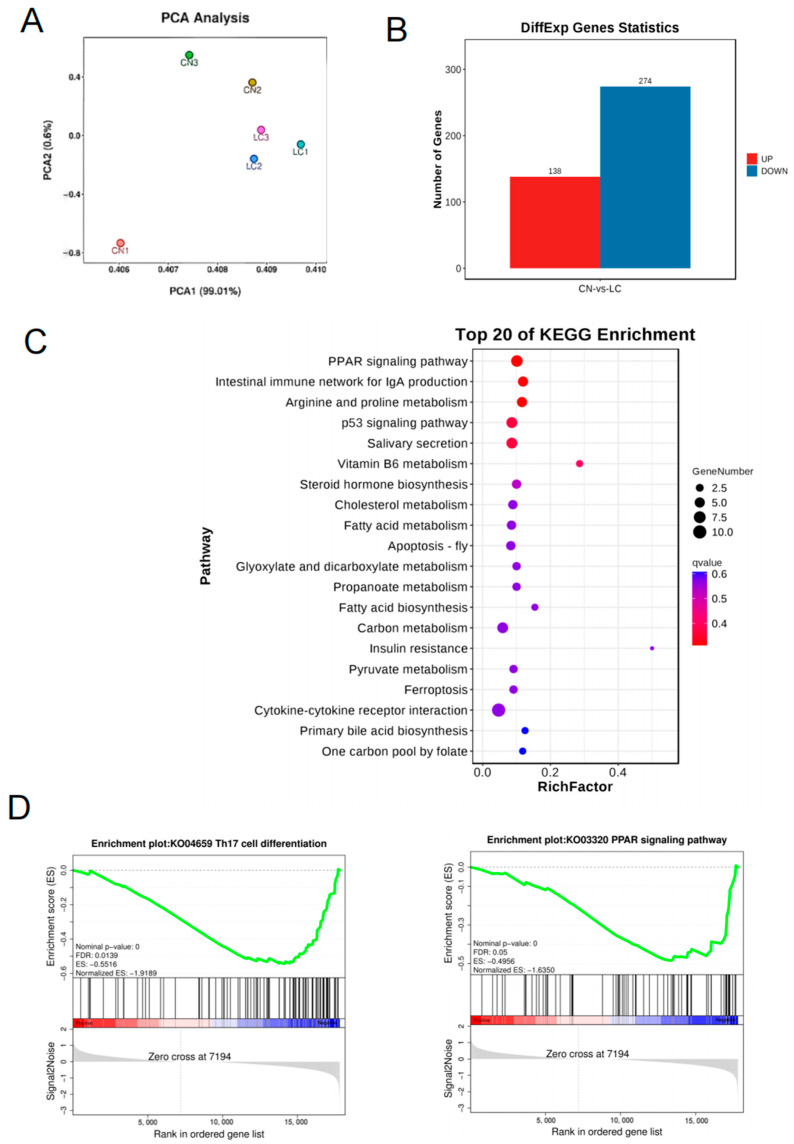
Functional enrichment analysis of the liver transcriptome between two groups. (**A**) Principal component analysis (PCA) based on whole transcripts in liver tissues between control (CN) and low-dose (LC) groups. (**B**) Statistics pertaining to differentially expressed genes. (**C**) Kyoto Encyclopedia of Genes and Genomes (KEGG) pathway enrichment analysis of differentially expressed (DiffExp) genes. (**D**) Gene set enrichment analysis plot for the Th17 cell differentiation and peroxisome proliferator-activated receptor (PPAR) signaling pathways.

**Table 1 animals-14-00920-t001:** Effects of dietary supplementation with different levels of VD_3_ on the growth performance of broilers.

Parameter	Groups			
	CN	LC	MC	HC
Initial BW (g)	32.26 ± 0.00	32.26 ± 0.00	32.26 ± 0.00	32.26 ± 0.00
Final BW (g)	1816.16 ± 29.19 ^b^	2109.49 ± 21.67 ^a^	2089.19 ± 18.32 ^a^	2090.28 ± 21.40 ^a^
ADFI (g)				
1 to 21 d	23.83 ± 0.26 ^b^	24.58 ± 0.11 ^a^	24.73 ± 0.15 ^a^	24.66 ± 0.09 ^a^
22 to 42 d	64.61 ± 0.82 ^b^	70.66 ± 0.68 ^a^	70.65 ± 0.67 ^a^	70.79 ± 1.00 ^a^
43 to 63 d	104.03 ± 2.50 ^b^	121.90 ± 1.75 ^a^	120.37 ± 1.57 ^a^	120.32 ± 1.50 ^a^
ADG (g)				
1 to 21 d	14.27 ± 0.17 ^b^	15.15 ± 0.11 ^a^	15.21 ± 0.11 ^a^	15.11 ± 0.09 ^a^
22 to 42 d	30.06 ± 0.35 ^b^	34.30 ± 0.40 ^a^	34.04 ± 0.39 ^a^	34.83 ± 0.22 ^a^
43 to 63 d	39.31 ± 1.55 ^b^	48.47 ± 0.75 ^a^	47.96 ± 0.69 ^a^	47.57 ± 0.51 ^a^
FCR (g/g)				
1 to 21 d	1.67 ± 0.00 ^a^	1.62 ± 0.00 ^c^	1.63 ± 0.00 ^bc^	1.64 ± 0.01 ^b^
22 to 42 d	2.15 ± 0.01 ^a^	2.06 ± 0.01 ^b^	2.08 ± 0.01 ^b^	2.06 ± 0.01 ^b^
43 to 63 d	2.66 ± 0.05 ^a^	2.52 ± 0.01 ^b^	2.51 ± 0.02 ^b^	2.53 ± 0.01 ^b^

Note: The values were calculated as the means ± standard error of the mean; different lowercase letters denote values that differ significantly *p* < 0.05. CN, control; LC, low concentration of VD_3_; MC, medium concentration of VD_3_; HC, high concentration of VD_3_; BW, body weight; ADG, average daily gain; ADFI, average daily feed intake; FCR, feed conversion ratio. Each value represents the mean of 6 replicates (n = 6).

**Table 2 animals-14-00920-t002:** Effects of different dietary levels of VD_3_ on the duodenal villus length and crypt depth of broilers.

Parameter	Groups			
	CN	LC	MC	HC
Villus length (μm)				
21 d	1479.82 ± 118.98	1626.17 ± 93.32	1557.59 ± 65.02	1556.09 ± 27.00
42 d	1947.45 ± 106.81	1998.59 ± 98.38	2143.83 ± 133.08	2034.90 ± 45.73
63 d	1385.86 ± 58.21 ^b^	1651.47 ± 44.81 ^ab^	1815.57 ± 151.21 ^a^	1919.24 ± 124.19 ^a^
Crypt depth (μm)				
21 d	227.82 ± 30.69	259.06 ± 13.94	205.66 ± 20.43	202.48 ± 9.16
42 d	317.92 ± 61.54	224.19 ± 56.07	258.54 ± 41.09	248.57 ± 48.14
63 d	287.86 ± 31.70 ^ab^	333.24 ± 19.83 ^a^	202.76 ± 27.39 ^c^	219.90 ± 24.40 ^bc^
V/C ratio				
21 d	6.92 ± 0.84	6.39 ± 0.25	7.90 ± 0.69	8.12 ± 0.38
42 d	5.67 ± 0.11	7.77 ± 1.17	8.66 ± 0.43	8.60 ± 0.81
63 d	5.48 ± 0.60 ^b^	5.07 ± 0.22 ^b^	9.41 ± 0.58 ^a^	9.09 ± 0.68 ^a^

Note: The values were calculated as the means ± standard error of the mean (n = 12); different lowercase letters denote values that differ significantly *p* < 0.05. CN, control; LC, low concentration of VD_3_; MC, medium concentration of VD_3_; HC, high concentration of VD_3_; V/C, villus length/crypt depth.

**Table 3 animals-14-00920-t003:** Effects of dietary supplementation with different levels of VD_3_ on the tibial characteristics of broilers.

Parameter	Groups			
	CN	LC	MC	HC
Weight (g)				
21 d	3.09 ± 0.06 ^b^	3.51 ± 0.10 ^a^	3.38 ± 0.12 ^a^	3.54 ± 0.05 ^a^
42 d	11.59 ± 0.19 ^b^	13.55 ± 0.35 ^a^	14.16 ± 0.28 ^a^	14.20 ± 0.27 ^a^
63 d	23.53 ± 0.79 ^b^	28.28 ± 0.42 ^a^	28.45 ± 0.53 ^a^	29.41 ± 0.59 ^a^
Length (mm)				
21 d	5.27 ± 0.06 ^b^	5.43 ± 0.07 ^ab^	5.42 ± 0.05 ^ab^	5.47 ± 0.05 ^a^
42 d	8.21 ± 0.06 ^b^	8.64 ± 0.07 ^a^	8.76 ± 0.05 ^a^	8.77 ± 0.09 ^a^
63 d	10.81 ± 0.14 ^b^	11.20 ± 0.09 ^a^	11.25 ± 0.07 ^a^	11.32 ± 0.11 ^a^
Tibial ash (%)				
21 d	40.17 ± 0.40 ^c^	44.27 ± 0.34 ^b^	44.38 ± 0.39 ^b^	45.67 ± 0.25 ^a^
42 d	43.14 ± 0.60 ^b^	44.97 ± 0.45 ^a^	45.45 ± 0.35 ^a^	45.22 ± 0.48 ^a^
63 d	47.46 ± 0.86 ^b^	49.96 ± 0.15 ^a^	49.81 ± 0.47 ^a^	49.69 ± 0.65 ^a^
Ca (%)				
21 d	15.37 ± 0.79 ^c^	18.43 ± 0.36 ^b^	21.28 ± 1.19 ^a^	20.04 ± 1.00 ^ab^
42 d	15.27 ± 0.59 ^b^	16.56 ± 0.094 ^ab^	16.74 ± 0.41 ^a^	16.30 ± 0.18 ^ab^
63 d	12.98 ± 0.91	11.48 ± 0.47	11.57 ± 0.89	12.23 ± 0.44
P (%)				
21 d	7.49 ± 0.39 ^c^	9.21 ± 0.16 ^bc^	12.90 ± 1.53 ^a^	10.89 ± 0.78 ^ab^
42 d	7.40 ± 0.61 ^b^	8.65 ± 0.14 ^a^	8.70 ± 0.32 ^a^	8.49 ± 0.09 ^a^
63 d	6.67 ± 0.63	5.47 ± 0.43	5.69 ± 0.47	6.20 ± 0.18

Note: The values were calculated as the means ± standard error of the mean (n = 12); different lowercase letters denote values that differ significantly *p* < 0.05. CN, control; LC, low concentration of VD_3_; MC, medium concentration of VD_3_; HC, high concentration of VD_3_.

## Data Availability

All data are contained in the manuscript.

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
