# Peer review of "The Effects of Optimal Dietary Vitamin D3 on Growth and Carcass Performance, Tibia Traits, Meat Quality, and Intestinal Morphology of Chinese Yellow-Feathered Broiler Chickens"

_animals, 2024, doi:10.3390/ani14060920_

Round 1

Reviewer 1 Report

Comments and Suggestions for Authors

Title: The effects of optimal dietary vitamin D3 on growth and carcass performance, tibia traits, meat quality, and intestinal morphology of Chinese yellow-feathered broiler chickens

The manuscript “The effects of optimal dietary vitamin D3 on growth and carcass performance, tibia traits, meat quality, and intestinal morphology of Chinese yellow-feathered broiler chickens” suggested dietary Vitamin D3 has positive effects on the growth, immunity, and bone development of broilers during the early stage, suggesting strategies to optimize poultry feeding. It is well-written article with some interesting findings; however, there are some corrections before its acceptance for publication:

Line 22: Mention the four treatments.

Line 30-31: Expand on the findings related to intestinal 16S rRNA sequencing and liver transcriptome analysis, highlighting the regulatory effects of dietary VD3 on lipid and immune-related metabolism.

Line 38: Consider adding a sentence at the beginning that introduces the specific focus of your study on yellow-feathered broilers.

Lines 45: Specify the deficiency risks faced by poultry in intensive farming conditions due to limited sunlight exposure.

Lines 63: Provide a brief explanation or rationale for why commercial broiler diets contain higher VD3 levels than recommended.

Lines 72: Specify the expected outcomes of your study more clearly to align with the title and research objectives.

Line 75: Specify the location of the chicken farm in Guangdong, China.

Lines 83: Consider including a brief rationale for the selection of these specific VD3 supplementation levels.

Lines 87: Describe the temperature and lighting conditions in more detail, including the specific ranges or settings used.

Lines 109-111: Provide more details on the measurements and procedures followed for recording various parameters after slaughtering the birds.

Lines 126-134: Describe the process of intestinal histomorphological measurements in more detail, including the staining procedure and measurement methods.

Lines 150-158: Explain the methodology for assessing meat quality, including the equipment used for pH and meat color measurements.

Line 214: Revise for clarity and conciseness: From day 1 to 63, the ADFI and ADG significantly increased in the LC, MC, and HC groups compared to the CN group (P < 0.05), while the FCR significantly decreased (P < 0.05).

Line 219: Specify the biochemical indices: Supplementation with VD3 had no substantial impact on the measured serum biochemical indices…

Line 378: Specify the mechanism: The improved growth performance induced by VD3 supplementation might act by promoting intestinal development and health.

Line 423-424: Specify the mechanism by which VD3 improves chicken meat color

Line 442: Specify the role of DNF00809, which is responsible for bone formation and differentiation through the microbiota-gut-metabolite-bone axis

Line 479-486: Overall, the conclusion could be strengthened by summarizing the key findings in each area (growth, carcass, bone, meat, and intestinal morphology) and emphasizing the practical implications for poultry producers. Additionally, considering the potential limitations of the study and suggesting directions for future research would further enhance the conclusion.

English grammar and sentence structure should be revised and corrected throughout the manuscript.

Comments on the Quality of English Language

English grammar and sentence structure should be revised and corrected throughout the manuscript.

Author Response

请参阅附件。

Reviewer 2 Report

Comments and Suggestions for Authors

Although there are many studies on adding vitamin D3 (VD3) in chickens, research on yellow-feathered broilers remains significant. This paper utilized a large sample size and systematically examined the experimental group, including carcass performance, tibia traits, meat quality, serum indicators, intestinal morphology, and liver transcriptome. However, there are several issues identified.

1. The paper divided the subjects into a control group and three treatment groups, with VD3 dosages of 1200/1000, 2400/2000, and 3600/3000 IU/kg, respectively. Compared to the control group, multiple production performance outcomes confirmed the positive effects of adding VD3, which was consistent with actual production results. However, as discussed by the authors on lines 372-373, the standard by the National Research Council (NRC) is 200 IU/kg, which is lower than the commercial addition levels for broilers, often found to be around 600 IU/kg in other literature. The study selected the LC group for further research on intestinal microbiota and liver transcriptome, considering it the best group. Typically, the optimal dosage in experiments is the middle dosage of the treatment groups, which was not the case in this study. Therefore, preliminary experiments should have been conducted to select appropriate dosages for more meaningful research outcomes.

2. What breed are the yellow-feathered broilers mentioned on line 76? Lines 97-99 describe sampling two chickens per pen without specifying whether they were male or female or one of each. The sex of the sampled chickens can significantly impact the results.

3. Table S1 does not clarify whether the nutritional components are analyzed or calculated values.

4. There are typographical errors in the titles of line 159 (2.7) and line 280 (3.5).

5. The significance description in line 224 and other tables' notes is inaccurate."

Comments on the Quality of English Language

Ok.

Reviewer 3 Report

Comments and Suggestions for Authors

REVIEW

Introduction to the topic of work

Recent studies have highlighted the pleiotropic function of vitamin D3 in terms of influence not only on calcium-phosphate, water-electrolyte and hormonal balance, but also on phenomena related to the proliferation and differentiation of cells belonging to the immune system, which seems to be closely related to the etipathogenesis of certain immunological diseases, cancer and allergies. The wide range of action of vitamin D3 is indicated by the widespread presence of its D-VDR receptor in various cells, tissues and organs of the body (heart, stomach, pancreas, brain, gonads, activated T and B lymphocytes).

Specific comments

Title

The title of the thesis reflects the scope of the research carried out.

Abstract

The abstract needs to be improved and supplemented with the most important results (figures) with indication of statistical significance.

Introduction to work

The introduction to the work is correctly written but needs to be completed. Please explain how vitamin D3 affects weight gain, feed intake and FCR.

Material and Methods

The experimental system as well as the analytical methods used are properly selected.

This chapter needs to be supplemented on the following points:

It is worth supplementing the methodology with the content of vitamin D3 in the base mixture. In what form was given vitamin D3.

I suggest we split up chapter two. 2. Measurements and sampling for the part concerning the assessment of fattening parameters, biochemical analysis of blood components and separately measurement of meat quality indicators.

The description of the molecular analysis part of the methodology is unqualified.

Results

The following data should be added to the description of the results:

It is worth supplementing the description with a section on the results of bird growth with numerical values.

It is worth supplementing the description with a section on the results of the characterisation of Ca, P and tibia in serum with numerical values.

It is worth supplementing the description with a section on the results of the characterisation of Ca, P and tibia in serum with numerical values of the indicators of carcass characteristics and meat quality

It is worth supplementing the description with a section on microbiota results with numerical values of individual bacterial groups.

Discussion

The chapter is correctly written, it should be supplemented with the following information:

Please refer directly to the results of the experiments on intestinal histology to those of other investigators.

Summary

In the summary, it should be clearly marked and given which dose of vitamin D3 is optimal based on the conducted studies.

Reviewer 4 Report

Comments and Suggestions for Authors

Review Report

Abstract

General Comments:

The abstract could be improved by starting with a brief introduction to the study. It mentions the number of broilers used in the study (n = 1440), but it does not provide any details about the experimental design, the treatment groups, how the broilers were divided into treatment groups, and the control groups used.

L25: Revised “improved intestinal morphology at the finisher stage” to improved intestinal morphology in the finisher stage.

L25-26: Revise “while the breast muscle meat color b* value decreased markedly under VD3 supplementation" to “while the b* value of breast muscle meat color decreased markedly under VD3 supplementation”.

L31-32: Revise “Dietary VD3 has positive effects on the growth…. To "had positive effects on the growth"

Introduction

L56: Revise “feed conversion rate” to “feed conversion ratio”.

L70-71: Revise “We hypothesized that” to “the hypothesis of the study was” ………….

Materials and methods

General comments:

The experimental design used is now well explained in the research. The authors just mentioned that birds were randomly assigned to four groups with six replicates per group, and each replicate contained 60 chickens without going further to indicate whether the design was a CRD, RCBD, or Factorial. Authors must clearly indicate the design used and the rationale behind the choice.

L78: Revise ….”60 chickens” to “60 chicks”

L95: Revise “Mortality was monitored on a daily basis, and any deceased birds were weighed” to “Mortality was monitored daily, and any deceased birds were weighed and used to adjust for feed intake.

L99: Insert “period” after “feed withdrawal”.

L139: Consider revising the statement to this “The surface of the bone was then dried using blotting paper."

Results

L211 Kindly delete “that in” after compared with.

L215: Revise “were increased in a statistically significant manner” to “increased significantly”…..

L231-232: Consider revising this statement “In detail, compared with those in the CN group, the VL and V/C ratio in the MC and HC groups had increased significantly by day 63 (P < 0.05), whereas there were no significant differences between the CN and LC groups (P > 0.05). Here is a suggestion “Specifically, on day 63, the VL and V/C ratio were significantly higher in the MC and HC groups compared to the CN group (P < 0.05). Conversely, no significant differences were observed between the CN and LC groups (P > 0.05).”

L267-268: Revise “Dietary supplementation with D3” to “Dietary supplementation of D3”.

L270-271: Delete “that” in the sentence “Compared with that the CN group”.

L342: Revise “We constructed three cDNA libraries from each group” to “Three cDNA libraries were constructed from each group”…..

L355: This statement “includin138 up- and 274 downregulated genes in the LC libraries” should be “including 138 up- and 274 downregulated genes in the LC libraries”.

Discussion

General Comments:

It would have been appropriate to have started the discussion with the growth parameters (i.e. feed intake, body weight, ADFI, ADG, and FCR). Again, that section was poorly discussed and needs to be improved.

L391: Kindly change “an higher” to “a higher”.

L466: Revise “metabolism-associated genes were significant enriched” to “metabolism-associated genes were significantly enriched”.

Conclusions

This section should be separated from the discussion.

References

Check the consistency of the referencing style. For instance, 3. “Mitchell, R. D.; Edwards, H. M. Jr.; McDaniel, G. R. (1997). The effects of ultraviolet light and cholecalciferol and its metabolites on the development of leg abnormalities in chickens genetically selected for a high and low incidence of tibial dyschondroplasia. Poultry science 1997, 76(2), 346–354” is at variance with the rest. Revise accordingly.

Comments on the Quality of English Language

The quality of the English is good and does not need any major improvement.

Round 2

Reviewer 1 Report

Comments and Suggestions for Authors

The manuscript is sufficiently improved and may be accepted in its present form for possible publication in Animals.

Reviewer 2 Report

Comments and Suggestions for Authors

The revised draft basically met the requirements.

Comments on the Quality of English Language

Ok.

Reviewer 3 Report

Comments and Suggestions for Authors

REVIEW 2

The authors of the article referred to most of the comments contained in the first version of the article.

The methodology was supplemented by adding the content of vitamin D3 in the base mixture. The section on growth results has been supplemented with numerical values.

Similarly, the section on the results of the characterisation of Ca, P and tibia in serum was supplemented with numerical values of the indices of carcass characteristics and meat quality. The section on microbiota results was supplemented with numerical values of individual bacterial groups.

The part of the chapter concerning the assessment of fattening parameters, biochemical analysis of blood components and, separately, the measurement of meat quality indicators has been separated.

The authors referred the experimental results on intestinal histological features to the results of scientific studies of other researchers.